# Stereoselective Synthesis of 1-Substituted Homotropanones, including Natural Alkaloid (−)-Adaline

**DOI:** 10.3390/molecules28052414

**Published:** 2023-03-06

**Authors:** Sandra Hernández-Ibáñez, Ana Sirvent, Miguel Yus, Francisco Foubelo

**Affiliations:** 1Departamento de Química Orgánica, Facultad de Ciencias, Universidad de Alicante, Apdo. 99, 03080 Alicante, Spain; 2Instituto de Síntesis Orgánica (ISO), Universidad de Alicante, Apdo. 99, 03080 Alicante, Spain; 3Centro de Innovación en Química Avanzada (ORFEO-CINQA), Universidad de Alicante, Apdo. 99, 03080 Alicante, Spain

**Keywords:** sulfinyl imines, decarboxylative Mannich reaction, cyclization, organocatalysis, homotropanone alkaloids

## Abstract

The stereocontrolled synthesis of 1-substituted homotropanones, using chiral *N*-*tert*-butanesulfinyl imines as reaction intermediates, is described. The reaction of organolithium and Grignard reagents with hydroxy Weinreb amides, chemoselective *N*-*tert*-butanesulfinyl aldimine formation from keto aldehydes, decarboxylative Mannich reaction with β-keto acids of these aldimines, and organocatalyzed L-proline intramolecular Mannich cyclization are key steps of this methodology. The utility of the method was demonstrated with a synthesis of the natural product (−)-adaline, and its enantiomer, (+)-adaline.

## 1. Introduction

Bicyclic alkaloids are significantly represented in nature and display a wide range of biological activities. Among them, compounds with a bridgehead nitrogen atom 1-azabicyclo[4.3.0]nonanes and 1-azabicyclo[4.4.0]decanes, the so-called indolizidines and quinolizidines [1], respectively, are more abundant than other azabicyclic systems with the nitrogen atom bonded to both bridgehead carbon atoms, such as 8-azabicyclo[3.2.1]octanes and 9-azabicyclo[3.3.1]nonanes, which are the basic skeletons of tropane [2] and homotropane alkaloids [3]. Representative tropane alkaloids include hyoscyamine, scopolamine, cocaine and calystegine A_3_, among others (Figure 1). Despite these alkaloids having a similar structure, they differ in biological activity, and have been used in the treatment of different diseases. For instance, hyoscyamine is an antimuscarinic, scopolamine is anticholinergic, calystegine A_3_ is employed to combat type 2 diabetes, and cocaine, an addictive stimulant drug, binds to the dopamine transporter, blocking the removal of dopamine from the synapse, producing an amplified signal to the receiving neurons. Homotropane alkaloids are less abundant than their homologous tropane derivatives, (+)-euphococcinine and (−)-adaline being the most significant (Figure 1). These compounds were found in the defensive fluid deployed by ladybirds to repel predators when disturbed. Precisely, (+)-euphococcinine was first isolated from the Australian coastal plant *Euphorbia atoto* [4] and was also found in the defense secretion of ladybirds *Cryptolaemus montrouzieri* [5] and Epilachna varivestis [6].

On the other hand, (−)-adaline was isolated from ladybird *Adalia bipunctata* [7] and *Cryptolaemus moutrouzieri* secretions [5]. It was recently found that (−)-adaline can target nicotinic acetylcholine receptors (nAChRs), acting as receptor antagonists, and also as an open channel blocker of nAChRs [8].

A wide variety of synthetic strategies have been employed in the synthesis of adaline and euphococcinine [3,9]. Recent approaches to these bicyclic systems are summarized in Figure 1. It should be mentioned that the Yu synthesis, starting from 3,4-dihydro-2-ethoxy-2*H*-pyran in a 6-step sequence, involves an intramolecular allylation of a cyclic imine as a key step [10]. The stereoselective synthesis of these alkaloids was also accomplished by Spino et al., taking advantage of *p*-menthane-3-carboxaldehyde as a chiral auxiliary. In the last step of this synthesis, the treatment with copper chloride of an isocyanate intermediate promotes an intramolecular conjugate addition of the nitrogen atom to the cyclic enone system [11]. A double cyclization through a four-step cascade reaction comprising *N*-desulfinylation, ketal hydrolysis, intramolecular imine formation and Mannich cyclization took place in the synthesis of (+)-adaline and (+)-euphococcinine reported by Davis and Edupuganti, upon treatment of a *N*-sulfinylamino ketone ketal with ammonium acetate and acetic acid [12]. More recently, Prasad and Khandare reported a four-step synthesis of these bicyclic homotropinone alkaloids, involving as key steps a diastereoselective addition of a Wittig phosphorene to a chiral *N*-*tert*-butanesulfinyl ketimine, a ring-closing metathesis, and a final intramolecular Michael reaction [13] (Figure 1).

The diastereoselective additions of different types of nucleophiles to chiral imines are recurrently used to access compounds with a nitrogen atom bonded to a stereogenic center. Of special relevance in this respect are *N*-*tert*-sulfinyl imines [14,15], in which the sulfinyl group plays a fundamental role in controlling the stereochemical pathway of these additions, which are highly stereospecific since the configuration of the sulfur atom determines the configuration of the newly formed stereocenter. Our group has studied in deep both the indium-mediated allylation [16] and the decarboxylative Mannich coupling of β-keto acids with *N*-*tert*-sulfinyl imines [17], and the application of the resulting homoallylamine derivatives [18,19,20,21] and β-amino ketones [22,23], respectively, to the synthesis of natural products.

Continuing our interest in the use of *N*-*tert*-butanesulfinyl imines as electrophiles [24,25,26,27,28], we decided to explore new synthetic pathways to access 1-substituted homotropanones in an enantioenriched form involving these chiral imines. Sequential decarboxylative Mannich reaction of a chiral *N*-*tert*-butanesulfinyl keto aldimine and a β-keto acid, followed by an organocatalyzed intramolecular Mannich reaction, upon desulfinylation, are key steps in the synthetic strategy we have envisioned for the synthesis of these bicyclic homotropanones, which is closely related to the strategy reported by Davis [12] (Figure 2).

## 2. Results and Discussion

### 2.1. Synthesis of N-tert-Butanesulfinyl Keto Aldimines ***5***

The synthesis of the target 1-substituted homotropanones started with the nucleophilic opening of δ-valerolactone (**1a**) upon treatment with *N,O*-dimethylhydroxylamine in the presence of trimethylaluminum in dichloromethane [29]. The corresponding Weinreb δ-hydroxyamide **2a** was formed in an almost quantitative yield. Further reaction with an excess of the corresponding organolithium or Grignard reagent led to the formation of δ-hydroxyketones **3** in excellent yields, in general. Structural diversity is introduced at this step of the here reported methodology. Only ketones **3f** and **3g** were obtained in moderate and low yields, respectively (55 and 36%). Phenylmagnesium bromide and benzylmagnesium chloride were the organometallic reagents involved in the synthesis of **3f** and **3g**, respectively (Figure 3).

Chiral *N*-*tert*-butanesulfinyl imino ketones **5** were key synthetic intermediates in the strategy depicted in Figure 2. They were prepared in a two-step process from hydroxyketones **3**. First, Swern oxidation of alcohols **3** produced keto aldehydes **4**. These reaction products were not isolated nor characterized. After the work-up, and removal of the solvents, the crude reaction mixtures were directly treated with (*S*)-*tert*-butanesulfinamide in the presence of titanium tetraethoxide, in THF, at room temperature. At this point, it merits mentioning that despite having two carbonyl groups (aldehyde and ketone) working under these reaction conditions, aldimines **5** were exclusively formed. It is known that more demanding reaction conditions are necessary for the synthesis of ketimines, which can be prepared with the same reagents and solvents, but working at higher temperatures, at least reflux of THF (Figure 3) [30]. Chiral sulfinyl imino ketones *ent*-**5d** and *ent*-**5e** were also prepared to work with (*R*)-*tert*-butanesulfinamide in the aldimine formation step. The expected reaction products were isolated after two synthetic operations in fairly good yields, ranging from 50 to 71% (Figure 3). 

### 2.2. Synthesis of N-tert-Butanesulfinyl Amino Diketones **6**

The next step in the proposed synthetic pathway to the target 9-azabicyclo[3.3.1]nonan-3-ones comprises a base-promoted decarboxylative-Mannich coupling of 3-oxobutanoic acid and *N*-*tert*-butanesulfinyl keto aldimines **5**. This methodology was developed in our research group [17], and we found that the resulting amino diketone derivatives were formed in a highly diastereoselective manner (>95:5 dr). We observed that yields were significantly improved when after 2 h of reaction, 1.5 equivalents of the base were added to the reaction mixture. Concerning the stereochemical pathway, the nucleophilic addition of the enolate took place to the *Re* face of the imine with *S*_S_ configuration. This result was rationalized considering an eight-membered cyclic transition state **A**, which was also supported in DFT calculations (Figure 3). In these transformations, yields ranged from 51 to 65%. Unfortunately, and after many attempts with varying reaction conditions, methyl ketone derivative **5a** led to the expected amino diketone **6a** in an extremely low yield. Only traces of this compound were detected from the ^1^H-NMR spectrum of the reaction crude (Figure 3). 

### 2.3. Synthesis of 1-Substituted 9-Azabicyclo[3.3.1]nonan-3-Ones ***7***

An intramolecular Mannich cyclization was the last step of the synthesis of compounds **7** from amino diketone derivatives **6**. We found that the treatment of compounds **6** first with a solution of hydrogen chloride in diethyl ether, in methanol as solvent, led to the free amine hydrochloride, which in a second step participated in a L-proline organocatalyzed intramolecular Mannich reaction, involving a six-membered cyclic imine initially formed, and the enolizable methyl group of the methyl ketone of the L-proline enamine. The intramolecular Mannich cyclization took place in a stereospecific manner, involving an iminium enamine carboxylate as a reaction intermediate, through, probably, a working model of type **B** depicted on Figure 3. The last step of this synthesis is rather similar to the one reported by Davis and Edupuganti in their synthesis of (+)-adaline and (+)-euphococcinine [12]. They found ammonium acetate and acetic acid at 75 °C the optimal conditions to transform a *N*-sulfinylamino ketone ketal into the corresponding 9-azabicyclo[3.3.1]nonan-3-one. When we applied these conditions to amino diketone derivative **6d**, natural product (−)-adaline **7d** was isolated in 63%, a lower yield than the one we found under the L-proline organocatalyzed cyclization (77%, Figure 3). The overall yield in this double cyclization transformation ranged from 55 to 77%. Importantly, we could prepare both enantiomers of these homotropanones by choosing the appropriate *tert*-butanesulfinamide enantiomer in the aldimine step formation, as it was exemplified in the synthesis of alkaloid (−)-adaline (**7d**) and its enantiomer (+)-adaline (*ent*-**7d**). 

The optical purity of compounds **7** was determined by GC using a column packet with a chiral stationary phase. Relatively high enantiomeric ratios were observed for compounds **7b**, **7c**, **7h** and **7i**. However, poorer enantioselectivities were found for pentyl and but-3-enyl substituted homotropanones **7d** and **7e** (Figure 3). Importantly, the starting amino diketone derivatives **6** were almost enantiopure (>95:5 dr), and it seems that the stereochemical integrity of these compounds was eroded in the double cyclization process. For that reason, and in order to rationalize the stereochemical outcome, we proposed the mechanism depicted in Figure 4. First, removing the *tert*-butanesulfinyl group was carried out under acidic conditions to produce the ammonium chloride **8**, which upon treatment with triethylamine and L-proline led to the iminium-enamine intermediate **9**. This compound is formed by a double condensation involving on one side the amine functionality and the carbonyl group, leading to the six-membered cyclic imine, and on the other side, the remaining carbonyl and the L-proline, forming the corresponding enamine. At this stage, isomerization could take place in some extension in the high polar reaction medium to give iminium **10**, losing the stereochemical integrity of the stereogenic center as a consequence. Intermediate **10** could be in equilibrium with iminium **9** and **9′**, which can participate in a second cyclization by nucleophilic attack of the enamine to the electrophilic carbon of the iminium through a transition state of type **B** (Figure 3), leading to the formation of homotropanones **7** and *ent*-**7**, respectively (Figure 4). 

### 2.4. Synthesis of 1-Substituted 8-Azabicyclo[3.2.1]octane ***16***

Taking advantage of the methodology developed for the synthesis of homotropanones **7**, we envisioned a synthesis under the same reaction conditions of more abundant natural tropanones, starting in this case from γ-butyrolactone **1b**. The formation of Weinreb amide **2b** from **1b** and hydroxy ketone **12** from **2b**, using *n*-buthyllithium, took place in high yields. Swern oxidation of alcohol **12**, followed by the formation of the imine **14**, proceeded in 68% overall yield. The next step was the base-promoted decarboxylative-Mannich coupling of 3-oxobutanoic acid and *N*-*tert*-butanesulfinyl keto aldimine **14**. The expected aminodiketone derivative **15** was isolated in 67% yield. However, the L-proline organocatalyzed intramolecular Mannich cyclization, under the reaction conditions that worked well for compounds **6**, did not provide tropanone derivative **16** (Figure 5). Complex reaction mixtures were always obtained even varying (solvent, temperature, stoichiometry) these optimal reaction conditions. 

## 3. Materials and Methods

### 3.1. General Information

Reagents and solvents were of reagent grade and purchased from commercial suppliers [Sigma-Aldridh (Saint Louis, MO, USA), Fisher Scientific (Kandel, Germany)] and used as received. (*S*)- and (*R*)-*tert*-butanesulfinamide were a gift from Medalchemy S.L. (Alicante, Spain) (>99% ee by chiral HPLC on a Chiracel AS column, 90:10 *n*-hexane/*i*-PrOH, 1.2 mL/min, λ = 222 nm).

Optical rotations were measured using a Jasco P-1030 polarimeter (Jasco, Tokyo, Japan) with a thermally jacketed 5 cm cell at approximately 23 °C, and concentrations (*c*) are given in g/100 mL. Low-resolution mass spectra (LRMS) were obtained in the electron impact mode (EI) with an Agilent MS5973N spectrometer with a SIS (Scientific Instrument Services) direct insertion probe (73DIP-1) at 70 eV and with an Agilent GC/MS5973N spectrometer in the electron impact mode (EI) at 70 eV. In both cases, fragment ions are given in *m*/*z* with relative intensities (%) in parentheses. High-resolution mass spectra (HRMS) were also carried out in the electron impact mode (EI) at 70 eV on an Agilent 7200 spectrometer equipped with a time of flight (TOF) analyzer, and the samples were introduced through a direct insertion probe or through an Agilent GC7890B (Agilent, Santa Clara, CA, USA). NMR spectra were recorded at 300 or 400 MHz for ^1^H NMR and at 75 or 100 MHz for ^13^C NMR with a Bruker AV300 Oxford or a Bruker AV400 spectrometers (Bruker, Karlsruhe, Germany), respectively, using CDCl_3_ as solvent, and TMS as internal standard (0.00 ppm). The data are reported as: (s = singlet, d = doublet, t = triplet, q = quartet, m = multiplet or unresolved, br s = broad signal, coupling constant(s) in Hz, integration). ^13^C NMR spectra were recorded with ^1^H-decoupling at 100 MHz and referenced to CDCl_3_ at 77.16 ppm. The DEPT-135 experiments were performed to assign CH, CH_2_ and CH_3_.

TLCs were performed on prefabricated Merck (Sigma-Aldrich, Saint Louis, MO, USA) aluminum plates with silica gel 60 coated with fluorescent indicator F_254_ and were visualized with phosphomolybdic acid (PMA) stain. The *R*_f_ values were calculated under these conditions. Flash chromatography was carried out on handpacked columns of silica gel 60 (230–400 mesh). GC-MS analysis were carried out in an Agilent 6890N spectrometer with FID detector, helium gas transportation (2 mL/min), injection pressure: 12 psi, temperature in detection with an injection blocks: 270 °C, column type HP-1 (12 m long, 0.22 mm internal diameter, 0.25 μm thickness methylsilicone rubber and OV-101 stationary phase). Temperature programs: (A) initial temperature (60 °C) for 3 min, heating 15 °C/min until final temperature (270 °C), final temperature (270 °C) for 10 min or (B) initial temperature (80 °C) for 5 min, heating 15 °C/min until final temperature (270 °C), final temperature (270 °C) for 10 min.

### 3.2. Preparation and Characterization of Compounds

#### 3.2.1. Synthesis of Weinreb’s Amides **2**

*General Procedure*. A solution of *N,O*-dimethylhydroxylamine hydrochloride (5.30 g, 54.0 mmol) in dry dichloromethane (30.0 mL) was stirred at −78 °C for 15 min. Then, a 2.0 M solution of AlMe_3_ in toluene (27.0 mL, 54.0 mmol) was slowly added over 1 h. The reaction mixture was stirred for 12 h and allowed to warm up until it reached 23 °C. After that, the reaction flask was cooled down at 0 °C, and the corresponding lactone **1** (18.0 mmol) was added. The resulting reaction mixture was stirred for 30 min at the same temperature, and for 2 h at 23 °C. After that, it was hydrolyzed with an aqueous solution (10.0 mL) of Rochelle’s salt (7.0 g). The resulting suspension was filtered through celite and repeatedly washed with dichloromethane (3 × 10 mL). Then, the solvent was removed under vacuum (15 Torr, <30 °C), giving rise to the expected compounds **2**, which was used in the next reaction step without the need for further purification.

5-Hydroxy-*N*-methoxy-*N*-methylpentanamide (**2a**) [31]. Following the general procedure, compound **2a** was obtained from δ-valerolactone (**1a**) as a colorless oil (2.870 g, 17.82 mmol, 99%): C_7_H_15_NO_3_; *R*_f_ 0.33 (hexane/EtOAc 1:4); ^1^H NMR (400 MHz, CDCl_3_) δ 3.68 (s, 3H), 3.63 (t, *J* = 6.3 Hz, 2H), 3.17 (s, 3H), 2.46 (t, *J* = 7.2 Hz, 2H), 1.78–1.66 (m, 2H), 1.66–1.52 (m, 2H); ^13^C NMR (100 MHz, CDCl_3_) δ 174.7 (C), 62.0 (CH_2_), 61.2 (CH_3_), 32.3 (CH_2_), 32.2 (CH_3_), 31.3 (CH_2_), 20.4 (CH_2_); LRMS (EI) *m*/*z* 162 (M^+^ + 1, 3%), 144 (3), 101 (91), 83 (51), 61 (99), 57 (27), 55 (100). 

4-Hydroxy-*N*-methoxy-*N*-methylbutanamide (**2b**) [32]. Following the general procedure, compound **2b** was obtained from γ-butyrolactone (**1b**) as a colorless oil (2.593 g, 17.64 mmol, 98%): C_6_H_13_NO_3_; *R*_f_ 0.36 (hexane/EtOAc 1:4); ^1^H NMR (400 MHz, CDCl_3_) δ 3.68 (s, 3H), 3.67 (t, *J* = 6.3 Hz, 3H), 3.17 (s, 3H), 2.57 (t, *J* = 7.0 Hz, 2H), 1.94–1.82 (m, 2H); ^13^C NMR (100 MHz, CDCl_3_) δ 174.7 (C), 62.1 (CH_2_), 61.2 (CH_3_), 32.12 (CH_3_), 28.9 (CH_2_), 27.3 (CH_2_); LRMS (EI) *m*/*z* 148 (M^+^ + 1, 1%), 87 (41), 69 (38), 61 (100).

#### 3.2.2. Synthesis of Hydroxy Ketones **3** and **12**

*General Procedure*. A solution of the corresponding Weireb’s amide **2** (6.0 mmol) in dry THF (50.0 mL) was stirred at −78 °C for 15 min. Then, a solution of the corresponding organolithium or Grignard reagent (24.0 mmol) was slowly added. The reaction mixture was stirred and allowed to warm up until it reached 23 °C for 15 h. After that, it was hydrolyzed with water (10 mL), extracted with ethyl acetate (3 × 20 mL), dried over magnesium sulfate, and the solvent was evaporated under a vacuum (15 Torr). The residue was pure enough to be used in the following step for compounds **3a**–**d** and **3i**. The residue for compounds **3e**–**h** and **12** was purified by column chromatography (silica gel, hexane/EtOAc) to yield pure products **3** and **12**.

6-Hydroxyhexan-2-one (**3a**). Following the general procedure, compound **3a** was obtained from amide **2a** as a colorless oil (689.1 mg, 5.94 mmol, 99%): C_6_H_12_O_2_; *R*_f_ 0.35 (hexane/EtOAc 1:1); ^1^H NMR (400 MHz, CDCl_3_) δ 3.62 (t, *J* = 6.3 Hz, 2H), 2.49 (t, *J* = 7.1 Hz, 2H), 2.15 (s, 3H), 1.74–1.61 (m, 3H), 1.61–1.49 (m, 2H); ^13^C NMR (100 MHz, CDCl_3_) δ 209.4 (C), 62.1, 43.2 (CH_2_), 32.0 (CH_2_), 29.8 (CH_3_), 19.8 (CH_2_); LRMS (EI) *m*/*z* 116 (M^+^, 1%), 98 (99), 83 (54), 58 (47), 56 (41), 55 (100); HRMS (EI-TOF) Calcd for C_6_H_12_O_2_ [M^+^]: 116.0884, found: 116.0889.

7-Hydroxyheptan-3-one (**3b**). Following the general procedure, compound **3b** was obtained from amide **2a** as a colorless oil (686.4 mg, 5.28 mmol, 88%): C_7_H_14_O_2_; *R*_f_ 0.40 (hexane/EtOAc 1:1); ^1^H NMR (400 MHz, CDCl_3_) δ 3.61 (t, *J* = 6.3 Hz, 2H), 2.48–2.40 (m, 4H), 1.70–1.50 (m, 5H), 1.04 (t, *J* = 7.2 Hz, 3H); ^13^C NMR (100 MHz, CDCl_3_) δ 212.1 (C), 62.1 (CH_2_), 41.9 (CH_2_), 35.9 (CH_2_), 32.1 (CH_2_), 19.9 (CH_2_), 7.8 (CH_3_); LRMS (EI) *m*/*z* 112 (M^+^-H_2_O, 27%), 101 (22), 83 (40), 57 (100), 55 (70); HRMS (EI-TOF) Calcd for C_7_H_14_O_2_ [M^+^]: 130.0994, found: 130.0990.

1-Hydroxynonan-5-one (**3c**). Following the general procedure, compound **3c** was obtained from amide **2a** as a colorless oil (929.1 mg, 5.88 mmol, 98%): C_9_H_18_O_2_; *R*_f_ 0.42 (hexane/EtOAc 1:1); ^1^H NMR (400 MHz, CDCl_3_) δ 3.62 (t, *J* = 6.2 Hz, 2H), 2.51–2.36 (m, 4H), 1.72–1.62 (m, 3H), 1.62–1.47 (m, 4H), 1.36–1.21 (m, 2H), 0.90 (t, *J* = 7.3 Hz, 3H); ^13^C NMR (100 MHz, CDCl_3_) δ 211.8 (C), 62.2 (CH_2_), 42.6 (CH_2_), 42.3 (CH_2_), 32.1 (CH_2_), 26.0 (CH_2_), 22.35 (CH_2_), 19.7 (CH_2_), 13.87 (CH_3_); LRMS (EI) *m*/*z* 140 (M^+^-H_2_O, 18%), 116 (32), 111 (16), 101 (29), 98 (70), 85 (100), 83 (63), 57 (91), 55 (85); HRMS (EI-TOF) Calcd for C_9_H_18_O_2_ [M^+^]: 158.1307, found: 158.1295.

1-Hydroxydecan-5-one (**3d**). Following the general procedure, compound **3d** was obtained from amide **2a** as a colorless oil (1.011 g, 5.88 mmol, 98%): C_10_H_20_O_2_; *R*_f_ 0.48 (hexane/EtOAc 1:1); ^1^H NMR (400 MHz, CDCl_3_) δ 3.63 (t, *J* = 6.1 Hz, 2H), 2.61–2.22 (m, 4H), 1.83 (s, 1H), 1.71–1.51 (m, 6H), 1.34–1.21 (m, 4H), 0.89 (t, *J* = 6.9 Hz, 3H); ^13^C NMR (100 MHz, CDCl_3_) δ 211.7 (C), 62.3 (CH_2_), 42.8 (CH_2_), 42.3 (CH_2_), 32.15 (CH_2_), 31.4 (CH_2_), 23.6 (CH_2_), 22.5 (CH_2_), 19.7 (CH_2_), 13.9 (CH_3_); LRMS (EI) *m*/*z* 154 (M^+^-H_2_O, 15%), 111 (24), 99 (100), 98 (90), 83 (89), 71 (85), 55 (94); HRMS (EI-TOF) Calcd for C_10_H_20_O_2_ [M^+^]: 172.1463, found: 172.1446.

9-Hydroxynon-1-en-5-one (**3e**). Following the general procedure, compound **3e** was obtained from amide **2a** as a colorless oil (655.2 mg, 4.20 mmol, 70%): C_9_H_16_O_2_; *R*_f_ 0.41 (hexane/EtOAc 1:1); ^1^H NMR (400 MHz, CDCl_3_) δ 5.82–5.77 (m, 1H), 5.12–4.81 (m, 2H), 3.61 (t, *J* = 6.2 Hz, 2H), 2.67–2.39 (m, 4H), 2.39–2.23 (m, 2H), 2.04 (s, 1H), 1.77–1.58 (m, 2H), 1.62–1.45 (m, 2H); ^13^C NMR (100 MHz, CDCl_3_) δ 210.6 (C), 137.2 (CH), 115.3 (CH_2_), 62.3 (CH_2_), 42.5 (CH_2_), 41.9 (CH_2_), 32.2 (CH_2_), 27.9 (CH_2_), 19.8 (CH_2_); LRMS (EI) *m*/*z* 138 (M^+^-H_2_O, 46%), 123 (18), 101 (23), 96 (27), 83 (52), 67 (18), 55 (100); HRMS (EI-TOF) Calcd for C_9_H_14_O [M^+^-H_2_O]: 138.1045, found: 138.1041.

5-Hydroxy-1-phenylpentan-1-one (**3f**). Following the general procedure, compound **3f** was obtained from amide **2a** as a colorless oil (587.4 mg, 3.30 mmol, 55%): C_11_H_14_O_2_; *R*_f_ 0.44 (hexane/EtOAc 1:1); ^1^H NMR (400 MHz, CDCl_3_) δ 8.06–7.91 (m, 2H), 7.67–7.50 (m, 1H), 7.52–7.41 (m, 2H), 3.79–3.63 (m, 2H), 3.05 (t, *J* = 7.1 Hz, 2H), 1.92–1.79 (m, 2H), 1.74 (s, 1H), 1.71–1.61 (m, 2H); ^13^C NMR (100 MHz, CDCl_3_) δ 200.5 (C), 137.2 (C), 133.1 (CH), 128.7 (CH), 128.2 (CH), 62.55 (CH_2_), 38.25 (CH_2_), 32.4 (CH_2_), 20.40 (CH_2_); LRMS (EI) *m*/*z* 178 (M^+^, 1%), 119 (20), 101 (45), 91 (40), 77 (100), 65 (25), 55 (65); HRMS (EI-TOF) Calcd for C_11_H_14_O_2_ [M^+^]: 178.0994, found: 178.0992.

7-Hydroxy-1-phenylheptan-3-one (**3h**). Following the general procedure, compound **3h** was obtained from amide **2a** as a colorless oil (848.4 mg, 4.20 mmol, 70%): C_13_H_18_O_2_; *R*_f_ 0.45 (hexane/EtOAc 1:1); ^1^H NMR (400 MHz, CDCl_3_) δ 7.39–7.13 (m, 5H), 3.61 (t, *J* = 6.3 Hz, 2H), 2.92 (t, *J* = 7.6 Hz, 2H), 2.45 (t, *J* = 7.1 Hz, 2H), 1.71–1.61 (m, 3H), 1.61–1.49 (m, 2H); ^13^C NMR (100 MHz, CDCl_3_) δ 210.3 (C), 141.1 (C), 128.5 (CH), 128.3 (CH), 126.1 (CH), 62.3 (CH_2_), 44.25 (CH_2_), 42.55 (CH_2_), 32.1 (CH_2_), 29.8 (CH_2_), 19.7 (CH_2_); LRMS (EI) *m*/*z* 206 (M^+^, 3%), 188 (54), 133 (37), 105 (75), 104 (28), 91 (100), 83 (19), 55 (22); HRMS (EI-TOF) Calcd for C_13_H_18_O_2_ [M^+^]: 206.1307, found: 206.1307.

8-Hydroxy-1-phenyloctan-4-one (**3i**). Following the general procedure, compound **3i** was obtained from amide **2a** as a colorless oil (1.095 g, 4.98 mmol, 83%): C_14_H_20_O_2_; *R*_f_ 0.50 (hexane/EtOAc 1:1); ^1^H NMR (400 MHz, CDCl_3_) δ 7.41–7.08 (m, 5H), 3.62 (t, *J* = 6.2 Hz, 2H), 2.69–2.56 (m, 2H), 2.49–2.36 (m, 3H), 2.02–1.84 (m, 4H), 1.79–1.45 (m, 4H); ^13^C NMR (100 MHz, CDCl_3_) δ 211.2 (C), 141.6 (C), 128.5 (CH), 128.5 (CH), 128.4 (CH), 126.0 (CH), 62.2 (CH_2_), 42.35 (CH_2_), 41.9 (CH_2_), 35.1 (CH_2_), 32.1 (CH_2_), 25.2 (CH_2_), 19.7 (CH_2_); LRMS (EI) *m*/*z* 202 (M^+^-H_2_O, 22%), 116 (20), 104 (44), 98 (100), 91 (38), 83 (12), 55 (13); HRMS (EI-TOF) Calcd for C_14_H_20_O_2_ [M^+^]: 220.1463, found: 220.1463.

1-Hydroxyoctan-4-one (**12**). Following the general procedure, compound **12** was obtained from amide **2b** as a colorless oil (795.0 mg, 5.52 mmol, 92%): C_8_H_16_O_2_; *R*_f_ 0.45 (hexane/EtOAc 1:1); ^1^H NMR (400 MHz, CDCl_3_) δ 3.64 (t, *J* = 6.1 Hz, 2H), 2.56 (t, *J* = 6.9 Hz, 2H), 2.43 (t, *J* = 7.5 Hz, 2H), 1.89–1.80 (m, 3H), 1.66–1.46 (m, 2H), 1.35–1.27 (m, 2H), 0.94–0.85 (m, 3H); ^13^C NMR (100 MHz, CDCl_3_) δ 212.2 (C), 62.15 (CH_2_), 42.7 (CH_2_), 39.5 (CH_2_), 26.6 (CH_2_), 26.0 (CH_2_), 22.4 (CH_2_), 13.89 (CH_3_); LRMS (EI) *m*/*z* 144 (M^+^, 2%), 126 (13), 102 (45), 97 (29), 87 (99), 85 (85), 69 (46), 58 (93), 57 (100); HRMS (EI-TOF) Calcd for C_8_H_16_O_2_ [M^+^]: 144.1150, found: 144.1153.

#### 3.2.3. Synthesis of *N*-*tert*-Butanesulfinyl Keto Aldimines **5** and **14**

*General Procedure*. A solution of oxalyl chloride (1.143 g, 0.772 mL, 9.0 mmol) in dry dichloromethane (20.0 mL) was stirred at −78 °C for 15 min. Then, DMSO (1.171 g, 1.065 mL, 15.0 mmol) was added dropwise. The reaction mixture was stirred for 5 min at the same temperature, and after that, a solution of the corresponding hydroxy ketone **3** or **12** (3.0 mmol) in dichloromethane (10.0 mL) was added. The resulting mixture was stirred for 15 min, and after that, Et_3_N (3.218 g, 4.432 mL, 31.8 mmol) was slowly added over 10 min. The reaction mixture was allowed to warm up and stirred for 2 h. Then, dichloromethane (20.0 mL) and a NH_4_Cl saturated aqueous solution (20.0 mL) were sequentially added. The aqueous layer was extracted with dichloromethane (3 × 25 mL), and the combined organic phases were washed with brine, dried over magnesium sulfate, and the solvent was evaporated under vacuum (15 Torr). The resulting keto aldehydes **4** were not isolated nor characterized and were used in the next step, the formation of the sulfinyl imine. A solution of the corresponding crude reaction mixture **4**, (*S*)- or (*R*)-*tert*-butanesulfinamide **1** (0.435 g, 3.6 mmol) and titanium tetraoxide (1.605 g, 1.507 mL, 7.2 mmol) in dry THF (5.0 mL) was stirred for 12 h at 23 °C. Then the resulting mixture was hydrolyzed with brine (2.0 mL), filtered through a celite pad, and washed with ethyl acetate (3 × 10 mL). The solvent was evaporated under vacuum (15 Torr), and the residue was purified by column chromatography (silica gel, hexane/EtOAc) to yield pure products **5** and **14**.

(*S*)-*N*-(5-Oxohex-1-ylidene)-*tert*-butanesulfinamide (**5a**). Following the general procedure, compound **5a** was obtained from hydroxy ketone **3a** as a yellow oil (462.4 mg, 2.13 mmol, 71%): C_10_H_19_NO_2_S; *R*_f_ 0.33 (hexane/EtOAc 3:1); [α]^20^_D_ +183.1 (*c* 1.03, CH_2_Cl_2_); ^1^H NMR (400 MHz, CDCl_3_) δ 8.07 (t, *J* = 4.3 Hz, 1H), 2.61–2.50 (m, 4H), 2.15 (s, 3H), 2.00–1.86 (m, 2H), 1.20 (s, 9H); ^13^C NMR (100 MHz, CDCl_3_) δ 207.9 (C), 168.8 (CH), 56.7 (C), 42.6 (CH_2_), 35.3 (CH_2_), 30.1 (CH_2_), 22.5 (CH_3_), 19.4 (CH_2_); LRMS (EI) *m*/*z* 160 (M^+^-C_4_H_9_, 4%), 112 (25), 103 (39), 70 (18), 57 (100), 55 (13), 43 (80), 41 (15); HRMS (EI-TOF) Calcd for C_6_H_10_NO_2_S [M^+^-C_4_H_9_] 160.0432, found 160.0422.

(*S*)-*N*-(5-Oxohep-1-ylidene)-*tert*-butanesulfinamide (**5b**). Following the general procedure, compound **5b** was obtained from hydroxy ketone **3b** as a yellow oil (416.0 mg, 1.80 mmol, 60%): C_11_H_21_NO_2_S; *R*_f_ 0.35 (hexane/EtOAc 3:1); [α]^20^_D_ +198.2 (*c* 0.84, CH_2_Cl_2_); ^1^H NMR (400 MHz, CDCl_3_) δ 8.07 (t, *J* = 4.3 Hz, 1H), 2.71–2.47 (m, 4H), 2.48–2.35 (m, 2H), 2.05–1.85 (m, 2H), 1.19 (s, 9H), 1.06 (t, *J* = 7.3 Hz, 3H); ^13^C NMR (100 MHz, CDCl_3_) δ 210.8 (C), 168.9 (CH), 56.7 (C), 41.3 (CH_2_), 36.1 (CH_2_), 35.4 (CH_2_), 22.4 (CH_3_), 19.45 (CH_2_), 7.1 (CH_3_); LRMS (EI) *m*/*z* 175 (M^+^-C_4_H_8_, 5%), 127 (25), 111 (23), 57 (100), 56 (10), 55 (11), 43 (13), 41 (22); HRMS (EI-TOF) Calcd for C_7_H_13_NO_2_S [M^+^-C_4_H_8_] 175.0667, found 175.0667.

(*S*)-*N*-(5-Oxonon-1-ylidene)-*tert*-butanesulfinamide (**5c**). Following the general procedure, compound **5c** was obtained from hydroxy ketone **3c** as a yellow oil (489.7 mg, 1.89 mmol, 63%): C_13_H_25_NO_2_S; *R*_f_ 0.40 (hexane/EtOAc 3:1); [α]^20^_D_ +131.6 (*c* 1.50, CH_2_Cl_2_); ^1^H NMR (400 MHz, CDCl_3_) δ 8.07 (t, *J* = 4.3 Hz, 1H), 2.59–2.47 (m, 4H), 2.40 (t, *J* = 7.4 Hz, 2H), 2.00–1.85 (m, 2H), 1.60–1.49 (m, 2H), 1.37–1.25 (m, 2H), 1.20 (s, 9H), 0.91 (t, *J* = 7.3 Hz, 3H); ^13^C NMR (100 MHz, CDCl_3_) δ 210.5 (C), 168.9 (CH), 56.7 (C), 42.8 (CH_2_), 41.7 (CH_2_), 35.4 (CH_2_), 26.1 (CH_2_), 22.5 (CH_3_), 19.5 (CH_2_), 14.0 (CH_3_); LRMS (EI) *m*/*z* 203 (M^+^-C_4_H_8_, 13%), 155 (75), 139 (12), 113 (17), 103 (14), 98 (14), 85 (13), 70 (14), 57 (100), 56 (24), 55 (15), 43 (14), 41 (32); HRMS (EI-TOF) Calcd for C_9_H_17_NO_2_S [M^+^-C_4_H_8_] 203.0980, found 203.0987.

(*S*)-*N*-(5-Oxodec-ylidene)-*tert*-butanesulfinamide (**5d**). Following the general procedure, compound **5d** was obtained from hydroxy ketone **3d** as a yellow oil (491.6 mg, 1.80 mmol, 60%): C_14_H_27_NO_2_S; *R*_f_ 0.42 (hexane/EtOAc 3:1); [α]^20^_D_ +153.6 (*c* 2.83, CH_2_Cl_2_); ^1^H NMR (400 MHz, CDCl_3_) δ 8.07 (t, *J* = 4.3 Hz, 1H), 2.62–2.45 (m, 4H), 2.39 (t, *J* = 7.4 Hz, 2H), 2.02–1.82 (m, 2H), 1.58 (s, 2H), 1.41–1.23 (m, 4H), 1.20 (s, 9H), 0.89 (t, *J* = 7.0 Hz, 3H); ^13^C NMR (100 MHz, CDCl_3_) δ 210.45 (C), 168.9 (CH), 56.7 (C), 43.0 (CH_2_), 41.7 (CH_2_), 35.4 (CH_2_), 31.5 (CH_2_), 23.7 (CH_2_), 22.55 (CH_3_), 19.4 (CH_2_), 14.0 (CH_3_); LRMS (EI) *m*/*z* 217 (M^+^-C_4_H_8_, 12%), 169 (75), 153 (15), 113 (12), 103 (16), 99 (15), 98 (18), 70 (19), 57 (100), 56 (33), 55 (20), 43 (34), 41 (32); HRMS (EI-TOF) Calcd for C_10_H_19_NO_2_S [M^+^-C_4_H_8_] 217.1136, found 217.1134.

(*R*)-*N*-(5-Oxodec-ylidene)-*tert*-butanesulfinamide (*ent*-**5d**). Following the general procedure, compound *ent*-**5d** was obtained from hydroxy ketone **3d** as a yellow oil (508.0 mg, 1.86 mmol, 62%). Physical and spectroscopic data were found to be the same as for **5d**. [α]^20^_D_ −183.1 (*c* 0.98, CH_2_Cl_2_).

(*S*)-*N*-(5-Oxonon-8-en-1-ylidene)-*tert*-butanesulfinamide (**5e**). Following the general procedure, compound **5e** was obtained from hydroxy ketone **3e** as a yellow oil (462.8 mg, 1.80 mmol, 60%): C_13_H_23_NO_2_S; *R*_f_ 0.41 (hexane/EtOAc 3:1); [α]^20^_D_ +181.3 (*c* 0.95, CH_2_Cl_2_); ^1^H NMR (400 MHz, CDCl_3_) δ 8.07 (t, *J* = 4.3 Hz, 1H), 5.87–5.67 (m, 1H), 5.07–4.89 (m, 2H), 2.60–2.44 (m, 6H), 2.40–2.26 (m, 2H), 2.02–1.86 (m, 2H), 1.19 (s, 9H); ^13^C NMR (100 MHz, CDCl_3_) δ 209.2 (C), 168.7 (CH), 136.95 (CH), 115.3 (CH_2_), 56.6 (C), 41.9 (CH_2_), 41.7 (CH_2_), 35.2 (CH_2_), 27.7 (CH_2_), 22.3 (CH_3_), 19.2 (CH_2_); LRMS (EI) *m*/*z* 201 (M^+^-C_4_H_8_, 21%), 153 (95), 137 (13), 111 (10), 103 (20), 99 (16), 98 (13), 96 (14), 57 (100), 56 (26), 55 (40), 41 (24); HRMS (EI-TOF) Calcd for C_9_H_15_NO_2_S [M^+^-C_4_H_8_] 201.0823, found 201.0823.

(*R*)-*N*-(5-Oxonon-8-en-1-ylidene)-*tert*-butanesulfinamide (*ent*-**5e**). Following the general procedure, compound *ent*-**5e** was obtained from hydroxy ketone **3e** as a yellow oil (555.3 mg, 2.16 mmol, 72%). Physical and spectroscopic data were found to be the same as for **5e**. [α]^20^_D_ −223.2 (*c* 1.02, CH_2_Cl_2_).

(*S*)-*N*-(5-Oxo-5-phenylpent-1-ylidene)-*tert*-butanesulfinamide (**5f**). Following the general procedure, compound **5f** was obtained from hydroxy ketone **3f** as a yellow oil (419.1 mg, 1.50 mmol, 50%): C_15_H_21_NO_2_S; *R*_f_ 0.33 (hexane/EtOAc 3:1); [α]^20^_D_ +164.7 (*c* 2.01, CH_2_Cl_2_); ^1^H NMR (400 MHz, CDCl_3_) δ 8.13 (t, *J* = 4.2 Hz, 1H), 8.03–7.90 (m, 2H), 7.64–7.54 (m, 1H), 7.54–7.42 (m, 2H), 3.18–3.01 (m, 2H), 2.70–2.60 (m, 2H), 2.20–2.02 (m, 2H), 1.20 (s, 9H); ^13^C NMR (100 MHz, CDCl_3_) δ 199.4 (C), 168.9 (CH), 136.9 (C), 133.3 (CH), 128.8 (CH), 128.1 (CH), 56.7 (C), 37.6 (CH_2_), 35.5 (CH_2_), 22.5 (CH_3_), 19.9 (CH_2_); LRMS (EI) *m*/*z* 223 (M^+^-C_4_H_8_, 14%), 176 (12), 175 (100), 133 (12), 105 (68), 103 (23), 77 (52), 73 (10), 70 (21), 57 (82), 56 (29), 51 (11), 43 (14), 41 (24); HRMS (EI-TOF) Calcd for C_11_H_13_NO_2_S [M^+^-C_4_H_8_] 223.0667, found 223.0665.

(*S*)-*N*-(5-Oxo-7-phenylhep-1-ylidene)-*tert*-butanesulfinamide (**5h**). Following the general procedure, compound **5h** was obtained from hydroxy ketone **3h** as a yellow oil (507.2 mg, 1.65 mmol, 55%): C_17_H_25_NO_2_S; *R*_f_ 0.35 (hexane/EtOAc 3:1); [α]^20^_D_ +195.5 (*c* 1.91, CH_2_Cl_2_); ^1^H NMR (400 MHz, CDCl_3_) δ 8.06 (t, *J* = 4.3 Hz, 1H), 7.34–7.25 (m, 2H), 7.25–7.15 (m, 3H), 2.92 (t, *J* = 7.6 Hz, 2H), 2.80–2.70 (m, 2H), 2.57–2.43 (m, 4H), 2.01–1.86 (m, 2H), 1.20 (s, 9H); ^13^C NMR (100 MHz, CDCl_3_) δ 209.25 (C), 168.8 (CH), 141.0 (C), 128.6 (CH), 128.45 (CH), 128.4 (CH), 126.3 (CH_2_), 56.72 (C), 44.5 (CH_2_), 42.0 (CH_2_), 35.3 (CH_2_), 29.9 (CH_2_), 22.5 (CH_3_), 19.3 (CH_2_); LRMS (EI) *m*/*z* 251 (M^+^-C_4_H_8_, 15%), 204 (14), 203 (94), 187 (13), 105 (36), 103 (21), 99 (25), 98 (15), 91 (91), 77 (10), 70 (12), 57 (100), 56 (32), 55 (15), 43 (11), 41 (23); HRMS (EI-TOF) Calcd for C_13_H_15_NO [M^+^-C_4_H_10_OS] 201.1154, found 201.1152.

(*S*)-*N*-(5-Oxo-8-phenyloct-1-ylidene)-*tert*-butanesulfinamide (**5i**). Following the general procedure, compound **5i** was obtained from hydroxy ketone **3i** as a yellow oil (588.1 mg, 1.83 mmol, 61%): C_18_H_27_NO_2_S; *R*_f_ 0.39 (hexane/EtOAc 3:1); [α]^20^_D_ +256.8 (*c* 1.93, CH_2_Cl_2_); ^1^H NMR (400 MHz, CDCl_3_) δ 8.07 (t, *J* = 4.3 Hz, 1H), 7.34–7.25 (m, 2H), 7.25–7.14 (m, 3H), 2.63 (t, *J* = 7.5 Hz, 2H), 2.58–2.49 (m, 2H), 2.50–2.44 (m, 2H), 2.46–2.38 (m, 2H), 1.99–1.83 (m, 4H), 1.20 (s, 9H); ^13^C NMR (100 MHz, CDCl_3_) δ 209.9 (C), 168.7 (CH), 141.50 (C), 128.5 (CH), 128.4 (CH), 126.0 (CH), 56.61 (C), 42.0 (CH_2_), 41.6 (CH_2_), 35.25 (CH_2_), 35.1 (CH_2_), 25.2 (CH_2_), 22.35 (CH_3_), 19.26 (CH_2_); LRMS (EI) *m*/*z* 265 (M^+^-C_4_H_8_, 19%), 218 (13), 217 (85), 201 (17), 147 (11), 126 (10), 113 (93), 105 (15), 104 (58), 103 (22), 98 (14), 91 (65), 70 (12), 57 (100), 56 (25), 55 (13), 41 (22); HRMS (EI-TOF) Calcd for C_14_H_19_NO_2_S [M^+^-C_4_H_8_] 265.1136, found 265.1128.

(*S*)-*N*-(4-Oxooct-1-ylidene)-*tert*-butanesulfinamide (**14**). Following the general procedure, compound **14** was obtained from hydroxy ketone **12** as a yellow oil (500.4 mg, 2.04 mmol, 68%): C_12_H_23_NO_2_S; *R*_f_ 0.33 (hexane/EtOAc 3:1); [α]^20^_D_ +124.3 (*c* 1.10, CH_2_Cl_2_); ^1^H NMR (400 MHz, CDCl_3_) δ 8.11 (t, *J* = 2.6 Hz, 1H), 2.96–2.63 (m, 4H), 2.52–2.38 (m, 2H), 1.61–1.47 (m, 2H), 1.37–1.25 (m, 2H), 1.15 (s, 9H), 0.91 (t, *J* = 7.3 Hz, 3H); ^13^C NMR (100 MHz, CDCl_3_) δ 208.8 (C), 167.8 (CH), 56.7 (C), 42.7 (CH_2_), 37.0 (CH_2_), 29.9 (CH_2_), 25.9 (CH_2_), 22.3 (CH_3_), 13.8 (CH_2_); LRMS (EI) *m*/*z* 189 (M^+^-C_4_H_8_, 11%), 141 (70), 127 (15), 99 (21), 98 (14), 83 (16) 57 (100), 56 (31), 55 (14), 41 (32); HRMS (EI-TOF) Calcd for C_8_H_14_NO_2_S [M^+^-C_4_H_9_] 188.0762, found 188.0753.

#### 3.2.4. Synthesis of *N*-*tert*-Butanesulfinyl Amino Diketones **6** and **15**

*General Procedure*. To a solution of 3-oxobutanoic acid (40.9 mg, 0.044 mL, 0.4 mmol) in THF (2.0 mL) at 0 °C was added a 1.0 M solution of LiOEt in THF (0.50 mL, 0.5 mmol). The reaction mixture was allowed to reach 10 °C, and the corresponding *N*-*tert*-butanesulfinyl keto aldimine **5** or **14** (0.2 mmol) was added, and stirring was continued for 2 h. If the starting imine **5** or **14** was not consumed after 2 h (TLC), a 1.0 M solution of LiOEt in THF (0.25 mL, 0.25 mmol) was added, and the resulting mixture was stirred for 12 h at 23 °C. After that, the reaction was hydrolyzed with water (10 mL), extracted with ethyl acetate (3 × 10 mL), dried over magnesium sulfate, and the solvent was evaporated (15 Torr). The residue was purified by column chromatography (silica gel, hexane/EtOAc) to yield pure products **6** and **15**.

(*S*_S_,*S*)-4-(*tert*-Butanesulfinylamino)decane-2,8-dione (**6b**). Following the general procedure, compound **6b** was obtained from keto aldimine **5b** as a yellow oil (32.4 mg, 0.112 mmol, 56%): C_14_H_27_NO_3_S; *R*_f_ 0.31 (hexane/EtOAc 1:3); [α]^20^_D_ +48.0 (*c* 0.38, CH_2_Cl_2_); ^1^H NMR (400 MHz, CDCl_3_) δ 3.98 (d, *J* = 9.0 Hz, 1H), 3.63–3.35 (m, 1H), 2.95–2.71 (m, 2H), 2.44–2.34 (m, 4H), 2.14 (s, 3H), 1.77–1.39 (m, 4H), 1.19 (s, 9H), 1.04 (t, *J* = 7.4 Hz, 3H); ^13^C NMR (100 MHz, CDCl_3_) δ 211.2 (C), 208.0 (C), 55.9 (C), 53.4 (CH), 48.9 (CH_2_), 41.6 (CH_2_), 35.95 (CH_2_), 35.0 (CH_2_), 31.0 (CH_3_), 22.6 (CH_3_), 20.25 (CH_2_), 7.8 (CH_3_); LRMS (EI) *m*/*z* 233 (M^+^-C_4_H_8_, 2%), 215 (19), 175 (11), 167 (19), 127 (100), 111 (11), 109 (10), 57 (74), 56 (14), 43 (35), 41 (16); HRMS (EI-TOF) Calcd for C_14_H_27_NO_3_S [M^+^] 289.1712, found 289.1699.

(*S*_S_,*S*)-4-(*tert*-Butanesulfinylamino)dodecane-2,8-dione (**6c**). Following the general procedure, compound **6c** was obtained from keto aldimine **5c** as a yellow oil (41.2 mg, 0.130 mmol, 65%): C_16_H_31_NO_2_S; *R*_f_ 0.36 (hexane/EtOAc 1:3); [α]^20^_D_ +47.3 (*c* 1.39, CH_2_Cl_2_); ^1^H NMR (400 MHz, CDCl_3_) δ 4.02 (d, *J* = 9.0 Hz, 1H), 3.52–3.48 (m, 1H), 3.05–2.66 (m, 2H), 2.47–2.33 (m, 4H), 2.15 (s, 3H), 1.64–1.44 (m, 5H), 1.38–1.22 (m, 3H), 1.21 (s, 9H), 0.90 (t, *J* = 7.3 Hz, 3H); ^13^C NMR (100 MHz, CDCl_3_) δ 211.1 (C), 208.3 (C), 56.1 (C), 53.6 (CH), 49.0 (CH_2_), 42.8 (CH_2_), 42.1 (CH_2_), 35.1 (CH_2_), 31.1 (CH_3_), 26.1 (CH_2_), 22.8 (CH_3_), 22.5 (CH_2_), 20.3 (CH_2_), 14.0 (CH_3_); LRMS (EI) *m*/*z* 261 (M^+^-C_4_H_8_, 2%), 243 (39), 221 (10), 203 (13), 195 (16), 156 (20), 155 (100), 153 (10), 147 (12), 139 (17), 113 (16), 97 (13), 85 (35), 73 (15), 70 (10), 57 (60), 56 (14), 43 (29), 41 (19); HRMS (EI-TOF) Calcd for C_12_H_23_NO_3_S [M^+^-C_4_H_8_] 261.1399, found 261.1384.

(*S*_S_,*S*)-4-(*tert*-Butanesulfinylamino)tridecane-2,8-dione (**6d**). Following the general procedure, compound **6d** was obtained from keto aldimine **5d** as a yellow oil (41.8 mg, 0.126 mmol, 63%): C_17_H_33_NO_3_S; *R*_f_ 0.39 (hexane/EtOAc 1:3); [α]^20^_D_ +47.1 (*c* 1.01, CH_2_Cl_2_); ^1^H NMR (400 MHz, CDCl_3_) δ 4.01 (d, *J* = 9.0 Hz, 1H), 3.53–3.49 (m, 1H), 2.97–2.73 (m, 2H), 2.46–2.32 (m, 4H), 2.15 (s, 3H), 1.67–1.41 (m, 5H), 1.44–1.21 (m, 5H), 1.21 (s, 9H), 0.89 (t, *J* = 7.1 Hz, 3H); ^13^C NMR (100 MHz, CDCl_3_) δ 210.95 (C), 208.1 (C), 55.9 (C), 53.4 (CH), 48.9 (CH_2_), 42.9 (CH_2_), 42.0 (CH_2_), 35.0 (CH_2_), 31.4 (CH_2_), 31.0 (CH_3_), 23.5 (CH_2_), 22.65 (CH_3_), 22.4 (CH_2_), 20.2 (CH_2_), 13.89 (CH_3_); LRMS (EI) *m*/*z* 275 (M^+^-C_4_H_8_, 1%), 257 (22), 170 (12), 169 (100), 153 (17), 99 (26), 71 (13), 57 (33), 56 (14), 43 (34), 41 (12); HRMS (EI-TOF) Calcd for C_17_H_33_NO_3_S [M^+^] 311.2181, found 331.2182.

(*R*_S_,*R*)-4-(*tert*-Butanesulfinylamino)tridecane-2,8-dione (*ent*-**6d**). Following the general procedure, compound *ent*-**6d** was obtained from keto aldimine *ent*-**5d** as a yellow oil (35.1 mg, 0.106 mmol, 53%). Physical and spectroscopic data were found to be the same as for **6d**. [α]^20^_D_ −49.3 (*c* 0.89, CH_2_Cl_2_).

(*S*_S_,*S*)-4-(*tert*-Butanesulfinylamino)dodec-11-ene-2,8-dione (**6e**). Following the general procedure, compound **6e** was obtained from keto aldimine **5e** as a yellow oil (34.7 mg, 0.110 mmol, 55%): C_16_H_29_NO_3_S; *R*_f_ 0.38 (hexane/EtOAc 1:3); [α]^20^_D_ +52.9 (*c* 0.52, CH_2_Cl_2_); ^1^H NMR (400 MHz, CDCl_3_) δ 5.79 (dd, *J* = 17.0, 10.3 Hz, 1H), 5.09–4.93 (m, 2H), 4.09 (d, *J* = 9.1 Hz, 1H), 3.56–3.44 (m, 1H), 2.85 (qd, *J* = 17.9, 5.2 Hz, 2H), 2.49 (t, *J* = 7.4 Hz, 2H), 2.42 (t, *J* = 6.8 Hz, 2H), 2.37–2.26 (m, 2H), 2.15 (s, 3H), 1.81–1.41 (m, 4H), 1.21 (s, 9H); ^13^C NMR (100 MHz, CDCl_3_) δ 209.9 (C), 208.2 (C), 137.1 (CH), 115.4 (CH_2_), 56.1 (C), 53.6 (CH), 49.0 (CH_2_), 42.2 (CH_2_), 41.95 (CH_2_), 35.1 (CH_2_), 31.1 (CH_3_), 27.85 (CH_2_), 22.8 (CH_3_), 20.3 (CH_2_); LRMS (EI) *m*/*z* 259 (M^+^-C_4_H_8_, 11%), 241 (42), 201 (11), 193 (19), 177 (54), 154 (11), 153 (86), 141 (29), 137 (15), 135 (17), 111 (14), 99 (11), 97 (17), 95 (23), 83 (100), 69 (11), 57 (86), 56 (26), 55 (84), 43 (48), 41 (30); HRMS (EI-TOF) Calcd for C_12_H_21_NO_3_S [M^+^-C_4_H_8_] 259.1242, found 259.1235.

(*R*_S_,*R*)-4-(*tert*-Butanesulfinylamino)dodec-11-ene-2,8-dione (*ent*-**6e**). Following the general procedure, compound *ent*-**6e** was obtained from keto aldimine *ent*-**5e** as a yellow oil (31.5 mg, 0.100 mmol, 50%). Physical and spectroscopic data were found to be the same as for **6e**. [α]^20^_D_ −54.6 (*c* 0.89, CH_2_Cl_2_).

(*S*_S_,*S*)-5-(*tert*-Butanesulfinylamino)-1-phenyloctane-1,7-dione (**6f**). Following the general procedure, compound **6f** was obtained from keto aldimine **5f** as a yellow oil (40.5 mg, 0.120 mmol, 60%): C_18_H_27_NO_3_S; *R*_f_ 0.31 (hexane/EtOAc 3:1); [α]^20^_D_ +46.2 (*c* 1.14, CH_2_Cl_2_); ^1^H NMR (400 MHz, CDCl_3_) δ 7.95–7.82 (m, 2H), 7.57–7.31 (m, 3H), 4.07–3.98 (m, 1H), 3.59–3.44 (m, 1H), 2.93 (t, *J* = 6.9 Hz, 2H), 2.84–2.80 (m, 2H), 2.09 (s, 3H), 1.94–1.44 (m, 4H), 1.14 (s, 9H); ^13^C NMR (100 MHz, CDCl_3_) δ 208.0 (C), 199.8 (C), 136.8 (C), 133.0 (CH), 128.6 (CH), 127.9 (CH), 55.9 (C), 53.4 (CH), 48.9 (CH_2_), 37.85 (CH_2_), 35.05 (CH_2_), 30.9 (CH_3_), 22.6 (CH_3_), 20.6 (CH_2_); LRMS (EI) *m*/*z* 281 (M^+^-C_4_H_8_, 5%), 199 (20), 175 (11), 105 (100), 77 (17), 57 (20), 43 (11); HRMS (EI-TOF) Calcd for C_14_H_19_NO_3_S [M^+^-C_4_H_8_] 281.1086, found 281.1076.

(*S*_S_,*S*)-4-(*tert*-Butanesulfinylamino)-10-phenyldecane-2,8-dione (**6h**). Following the general procedure, compound **6h** was obtained from keto aldimine **5h** as a yellow oil (44.6 mg, 0.122 mmol, 61%): C_20_H_31_NO_3_S; *R*_f_ 0.34 (hexane/EtOAc 1:3); [α]^20^_D_ +39.7 (*c* 0.58, CH_2_Cl_2_); ^1^H NMR (400 MHz, CDCl_3_) δ 7.36–7.06 (m, 5H), 4.03 (d, *J* = 9.1 Hz, 1H), 3.48 (d, *J* = 4.8 Hz, 1H), 2.93–2.87 (m, 2H), 2.91–2.74 (m, 2H), 2.75–2.66 (m, 2H), 2.40 (t, *J* = 6.8 Hz, 2H), 2.15 (s, 3H), 1.79–1.59 (m, 2H), 1.62–1.48 (m, 2H), 1.20 (s, 9H); ^13^C NMR (100 MHz, CDCl_3_) δ 209.7 (C), 208.1 (C), 141.0 (C), 128.5 (CH), 128.3 (CH), 126.1 (CH), 56.0 (C), 53.45 (CH), 48.9 (CH_2_), 44.3 (CH_2_), 42.3 (CH_2_), 34.9 (CH_2_), 31.0 (CH_3_), 29.7 (CH_2_), 22.7 (CH_3_), 20.15 (CH_2_); LRMS (EI) *m*/*z* 292 (M^+^-C_4_H_9_O, 6%), 291 (30), 243 (19), 204 (16), 203 (100), 187 (14), 133 (19), 105 (44), 103 (10), 99 (15), 97 (13), 91 (74), 57 (51), 56 (19), 43 (35), 41 (16); HRMS (EI-TOF) Calcd for C_16_H_22_NO_3_S [M^+^-C_4_H_8_] 308.1320, found 308.1324.

(*S*_S_,*S*)-4-(*tert*-Butanesulfinylamino)-11-phenylundecane-2,8-dione (**6i**). Following the general procedure, compound **6i** was obtained from ketoaldimine **5i** as a yellow oil (38.7 mg, 0.102 mmol, 51%): C_21_H_33_NO_3_S; *R*_f_ 0.35 (hexane/EtOAc 1:3); [α]^20^_D_ +46.8 (*c* 1.49, CH_2_Cl_2_); ^1^H NMR (400 MHz, CDCl_3_) δ 7.37–7.06 (m, 5H), 3.99 (d, *J* = 9.1 Hz, 1H), 3.55–3.40 (m, 1H), 2.97–2.74 (m, 2H), 2.61 (t, *J* = 7.6 Hz, 2H), 2.46–2.33 (m, 4H), 2.14 (s, 3H), 1.98–1.82 (m, 2H), 1.75–1.38 (m, 4H), 1.20 (s, 9H); ^13^C NMR (100 MHz, CDCl_3_) δ 210.4 (C), 208.1 (C), 141.5 (C), 128.5 (CH), 128.4 (CH), 126.0 (CH), 55.9 (C), 53.4 (CH), 48.9 (CH_2_), 42.1 (CH_2_), 42.0 (CH_2_), 35.1 (CH_2_), 35.0 (CH_2_), 31.0 (CH_3_), 25.2 (CH_2_), 22.7 (CH_3_), 20.2 (CH_2_); LRMS (EI) *m*/*z* 323 (M^+^-C_4_H_8_, 1%), 305 (51), 218 (18), 217 (100), 201 (26), 153 (27), 147 (60), 113 (78), 105 (20), 104 (67), 91 (31), 91 (89), 70 (15), 57 (92), 56 (27), 43 (66), 41 (33); HRMS (EI-TOF) Calcd for C_17_H_24_NO_3_S [M^+^-C_4_H_8_] 322.1520, found 322.1524.

(*S*_S_,*S*)-4-(*tert*-Butanesulfinylamino)undecane-2,7-dione (**15**). Following the general procedure, compound **15** was obtained from keto aldimine **14** as a yellow oil (40.6 mg, 0.134 mmol, 67%): C_15_H_29_NO_2_S; *R*_f_ 0.29 (hexane/EtOAc 1:3); [α]^20^_D_ +39.3 (*c* 0.80, CH_2_Cl_2_); ^1^H NMR (400 MHz, CDCl_3_) δ 4.05 (d, *J* = 9.6 Hz, 1H), 3.61–3.38 (m, 1H), 2.88 (qd, *J* = 17.9, 5.0 Hz, 2H), 2.63–2.49 (m, 2H), 2.44–2.34 (m, 2H), 2.16 (s, 3H), 1.89–1.77 (m, 2H), 1.64–1.47 (m, 2H), 1.33–1.26 (m, 2H), 1.22 (s, 9H), 0.90 (t, *J* = 7.3 Hz, 3H); ^13^C NMR (100 MHz, CDCl_3_) δ 210.7 (C), 208.2 (C), 56.1 (C), 53.6 (CH), 49.4 (CH_2_), 42.8 (CH_2_), 39.45 (CH_2_), 31.1 (CH_3_), 29.4 (CH_2_), 26.1 (CH_2_), 22.8 (CH_3_), 22.4 (CH_2_), 13.9 (CH_3_); LRMS (EI) *m*/*z* 247 (M^+^-C_4_H_8_, 8%), 204 (12), 198 (25), 155 (26), 140 (20), 113 (28), 111 (17), 105 (100), 57 (80), 56 (18), 43 (19); HRMS (EI-TOF) Calcd for C_11_H_19_NO_3_S [M^+^-C_4_H_10_] 245.1086, found 245.1086.

#### 3.2.5. Synthesis of 1-Substituted 9-Azabicyclo[3.3.1]nonan-3-ones **7**

*General Procedure*. To a solution of the corresponding amino diketone derivative **6** (0.2 mmol) in MeOH (2.0 mL), a 2.0 M solution of HCl was added into Et_2_O (1.0 mL, 2.0 mmol) at 0 °C. After stirring for 30 min at this temperature, the disappearance of the starting reagent and the subsequent formation of the ammonium chloride was monitored by TLC. After that, the solvents were removed under vacuum (15 Torr), and EtOH, 2.0 mL), triethylamine (20.3 g, 0.028 mL, 0.2 mmol), L-proline (4.6 mg, 0.04 mmol) and anhydrous magnesium sulfate (24.0 mg, 0.2 mmol) were successively added to the flask containing the dry ammonium salt, and the resulting reaction mixture was stirred for 12 h at 23 °C. After that, the reaction was hydrolyzed with a sodium bicarbonate saturated aqueous solution (10 mL), extracted with dichloromethane (3 × 10 mL), dried over magnesium sulfate, and the solvents were evaporated (15 Torr). The residue was purified by column chromatography (silica gel, hexane/EtOAc) to yield pure products **7**.

(1*R*,5*S*)-1-Ethyl-9-azabicyclo[3.3.1]nonan-3-one (**7b**). Following the general procedure, compound **7b** was obtained from amino diketone derivative **6b** as a yellow oil (25.1 mg, 0.150 mmol, 75%): C_10_H_17_NO; 2.5:97.5 er [GC (CP-Chirasil-Dex CB column, T_inlet_ = 275 °C, T_detector_ = 250 °C, T_column_ = 70 °C and 70–200 °C (4 °C/min), *p* = 101 kPa): t_minor_ = 26.75 min, t_major_ = 26.89 min]; *R*_f_ 0.55 (CH_2_Cl_2_/EtOH 10:1); [α]^20^_D_ −5.6 (*c* 1.23, CH_2_Cl_2_); ^1^H NMR (400 MHz, CDCl_3_) δ 3.78 (s, 1H), 2.70 (dd, *J* = 16.3, 6.8 Hz, 1H), 2.43 (d, *J* = 16.9 Hz, 2H), 2.30 (d, *J* = 16.5 Hz, 1H), 1.94–1.77 (m, 1H), 1.77–1.35 (m, 6H), 1.22–1.08 (m, 2H), 0.94 (t, *J* = 7.6 Hz, 3H); ^13^C NMR (100 MHz, CDCl_3_) δ 210.2 (C), 55.6 (C), 50.6 (CH_2_), 49.9 (CH), 46.2 (CH_2_), 36.7 (CH_2_), 35.7 (CH_2_), 31.2 (CH_2_), 17.70 (CH_2_), 7.15 (CH_3_); LRMS (EI) *m*/*z* 167 (M^+^, 43%), 125 (31), 124 (100), 110 (53), 96 (33), 82 (20); HRMS (EI-TOF) Calcd for C_10_H_17_NO [M^+^] 167.1310, found 167.1311.

(1*R*,5*S*)-1-Butyl-9-azabicyclo[3.3.1]nonan-3-one (**7c**). Following the general procedure, compound **7c** was obtained from amino diketone derivative **6c** as a yellow oil (27.4 mg, 0.140 mmol, 70%): C_12_H_21_NO; 93:7 er [GC (CP-Chirasil-Dex CB column, T_inlet_ = 275 °C, T_detector_ = 250 °C, T_column_ = 70 °C and 70–200 °C (4 °C/min), *p* = 101 kPa): t_major_ = 35.84 min, t_minor_ = 36.64 min]; *R*_f_ 0.56 (CH_2_Cl_2_/EtOH 10:1); [α]^20^_D_ −8.0 (*c* 0.51, CH_2_Cl_2_); ^1^H NMR (400 MHz, CDCl_3_) δ 3.87–3.69 (m, 2H), 2.72 (dd, *J* = 16.4, 6.8 Hz, 1H), 2.50–2.30 (m, 3H), 1.73–1.47 (m, 6H), 1.38–1.21 (m, 6H), 0.99–0.87 (m, 3H); ^13^C NMR (100 MHz, CDCl_3_) δ 209.8 (C), 55.4 (C), 50.7 (CH_2_), 49.5 (CH), 45.7 (CH_2_), 43.6 (CH_2_), 35.8 (CH_2_), 30.75 (CH_2_), 24.7 (CH_2_), 23.1 (CH_2_), 17.45 (CH_2_), 14.0 (CH_3_); LRMS (EI) *m*/*z* 195 (M^+^, 21%), 166 (22), 153 (100), 152 (77), 138 (33), 110 (58), 96 (96); HRMS (EI-TOF) Calcd for C_12_H_21_NO [M^+^] 195.1623, found 195.1633.

(1*R*,5*S*)-1-Pentyl-9-azabicyclo[3.3.1]nonan-3-one, (−)-Adaline (**7d**). Following the general procedure, compound **7d** was obtained from amino diketone derivative **6d** as a yellow oil (32.2 mg, 0.154 mmol, 77%): C_13_H_23_NO; 86:14 er [GC (CP-Chirasil-Dex CB column, T_inlet_ = 275 °C, T_detector_ = 250 °C, T_column_ = 70 °C and 70–200 °C (4 °C/min), *p* = 101 kPa): t_major_ = 38.21 min, t_minor_ = 38.90 min]; *R*_f_ 0.57 (CH_2_Cl_2_/EtOH 10:1); [α]^20^_D_ −8.6 (*c* 1.49, CH_2_Cl_2_) [lit. [33] −11.4 (*c* 0.74, CHCl_3_)]; ^1^H NMR (400 MHz, CDCl_3_) δ 3.74 (s, 1H), 2.65 (dd, *J* = 16.2, 6.7 Hz, 1H), 2.49–2.34 (m, 2H), 2.29 (d, *J* = 16.1 Hz, 1H), 1.82–1.13 (m, 15H), 0.95–0.84 (m, 3H); ^13^C NMR (100 MHz, CDCl_3_) δ 210.3 (C), 55.3 (C), 51.0 (CH_2_), 49.6 (CH), 46.0 (CH_2_), 44.1 (CH_2_), 36.1 (CH_2_), 32.4 (CH_2_), 31.0 (CH_2_), 22.6 (CH_2_), 22.3 (CH_2_), 17.65 (CH_2_), 14.1 (CH_3_); LRMS (EI) *m*/*z* 209 (M^+^, 13%), 180 (17), 166 (99), 153 (87), 138 (21), 124 (16), 110 (53), 96 (100); HRMS (EI-TOF) Calcd for C_13_H_23_NO [M^+^] 209.1780, found 209.1777.

(1*S*,5*R*)-1-Pentyl-9-azabicyclo[3.3.1]nonan-3-one, (+)-Adaline (*ent*-**7d**). Following the general procedure, compound *ent*-**7b** was obtained from amino diketone derivative *ent*-**6d** as a yellow oil (31.8 mg, 0.152 mmol, 76%). Physical and spectroscopic data were found to be the same as for **7d**. 23:77 er [GC (CP-Chirasil-Dex CB column, T_inlet_ = 275 °C, T_detector_ = 250 °C, T_column_ = 70 °C and 70–200 °C (4 °C/min), *p* = 101 kPa): t_minor_ = 38.50 min, t_major_ = 38.87 min]; [α]^20^_D_ +8.0 (*c* 2.70, CH_2_Cl_2_).

(1*S*,5*S*)-1-(But-3-en-1-yl)-9-azabicyclo[3.3.1]nonan-3-one (**7e**). Following the general procedure, compound **7e** was obtained from amino diketone derivative **6e** as a yellow oil (21.3 mg, 0.110 mmol, 55%): C_12_H_19_NO; 88:12 er [GC (CP-Chirasil-Dex CB column, T_inlet_ = 275 °C, T_detector_ = 250 °C, T_column_ = 70 °C and 70–200 °C (4 °C/min), *p* = 101 kPa): t_major_ = 36.31 min, t_minor_ = 37.02 min]; *R*_f_ 0.55 (CH_2_Cl_2_/EtOH 10:1); [α]^20^_D_ −8.5 (*c* 1.25, CH_2_Cl_2_); ^1^H NMR (400 MHz, CDCl_3_) δ 5.90–5.70 (m, 1H), 5.15–4.89 (m, 2H), 4.71–4.67 (m, 2H), 3.80 (t, *J* = 5.3 Hz, 1H), 2.84–2.60 (m, 1H), 2.55–2.31 (m, 3H), 2.20–2.06 (m, 2H), 1.78–1.46 (m, 7H); ^13^C NMR (100 MHz, CDCl_3_) δ 209.4 (C), 138.0 (CH), 115.3 (CH_2_), 55.5 (C), 50.7 (CH_2_), 49.4 (CH), 45.6 (CH_2_), 42.75 (CH_2_), 35.8 (CH_2_), 30.7 (CH_2_), 27.1 (CH_2_), 17.5 (CH_2_); LRMS (EI) *m*/*z* 193 (M^+^, 11%), 150 (100), 136 (93), 122 (35), 110 (44), 96 (51), 82 (38), 67 (22), 55 (39); HRMS (EI-TOF) Calcd for C_12_H_19_NO [M^+^] 193.1467, found 193.1471.

(1*R*,5*R*)-1-(But-3-en-1-yl)-9-azabicyclo[3.3.1]nonan-3-one (*ent*-**7e**). Following the general procedure, compound *ent*-**7e** was obtained from amino diketone derivative *ent*-**6e** as a yellow oil (20.1 mg, 0.104 mmol, 52%). Physical and spectroscopic data were found to be the same as for **7e**. 19.5:80.5 er [GC (CP-Chirasil-Dex CB column, T_inlet_ = 275 °C, T_detector_ = 250 °C, T_column_ = 70 °C and 70–200 °C (4 °C/min), *p* = 101 kPa): t_minor_ = 36.61 min, t_major_ = 37.06 min]; [α]^20^_D_ + 7.9 (*c* 1.57, CH_2_Cl_2_).

(1*S*,5*S*)-1-(2-Phenylethyl)-9-azabicyclo[3.3.1]nonan-3-one (**7h**). Following the general procedure, compound **7h** was obtained from amino diketone derivative **6h** as a yellow oil (30.2 mg, 0.124 mmol, 62%): C_16_H_21_NO; 92.5:7.5 er [GC (CP-Chirasil-Dex CB column, T_inlet_ = 275 °C, T_detector_ = 250 °C, T_column_ = 70 °C and 70–200 °C (4 °C/min), *p* = 101 kPa): t_major_ = 53.44 min, t_minor_ = 53.70 min]; *R*_f_ 0.58 (CH_2_Cl_2_/EtOH 10:1); [α]^20^_D_ −5.5 (*c* 1.70, CH_2_Cl_2_); ^1^H NMR (400 MHz, CDCl_3_) δ 7.43–7.03 (m, 5H), 3.73 (s, 1H), 2.78–2.09 (m, 7H), 1.86–1.43 (m, 8H); ^13^C NMR (100 MHz, CDCl_3_) δ 210.5 (C), 141.9 (C), 128.7 (CH), 128.4 (CH), 126.15 (CH), 55.2 (C), 51.4 (CH_2_), 49.75 (CH), 46.4 (CH_2_), 46.3 (CH_2_), 36.4 (CH_2_), 31.35 (CH_2_), 29.3 (CH_2_), 17.81 (CH_2_); LRMS (EI) *m*/*z* 243 (M^+^, 73%), 200 (47), 186 (100), 184 (42), 91 (61); HRMS (EI-TOF) Calcd for C_16_H_21_NO [M^+^] 243.1623, found 243.1621.

(1*R*,5*S*)-1-(3-Phenylpropyl)-9-azabicyclo[3.3.1]nonan-3-one (**7i**). Following the general procedure, compound **7i** was obtained from amino diketone derivative **6i** as a yellow oil (34.5 mg, 0.134 mmol, 67%): C_17_H_23_NO; 91.3:8.7 er [GC (CP-Chirasil-Dex CB column, T_inlet_ = 275 °C, T_detector_ = 250 °C, T_column_ = 100 °C and 100–200 °C (4 °C/min), *p* = 101 kPa): t_major_ = 66.97 min, t_minor_ = 67.30 min]; *R*_f_ 0.61 (CH_2_Cl_2_/EtOH 10:1); [α]^20^_D_ −7.8 (*c* 0.69, CH_2_Cl_2_); ^1^H NMR (400 MHz, CDCl_3_) δ 7.42–7.02 (m, 5H), 3.82–3.59 (m, 1H), 2.68–2.12 (m, 7H), 1.76–1.35 (m, 10H); ^13^C NMR (100 MHz, CDCl_3_) δ 211.1 (C), 142.0 (C), 128.45 (CH), 125.97 (CH), 54.75 (C), 51.5 (CH_2_), 49.7 (CH), 46.6 (CH_2_), 44.1 (CH_2_), 36.5 (CH_2_), 36.3 (CH_2_), 31.5 (CH_2_), 24.7 (CH_2_), 17.8 (CH_2_); LRMS (EI) *m*/*z* 257 (M^+^, 5%), 214 (24), 154 (10), 153 (100), 110 (22), 96 (89), 91 (20); HRMS (EI-TOF) Calcd for C_17_H_23_NO [M^+^] 257.1780, found 257.1780.

Copies of ^1^H-NMR, ^13^C-NMR, DEPT spectra of compounds **5**, **6**, **7**, **14** and **15**, as well as chiral GC chromatograms of compounds **7** are available is Appendix A.

## 4. Conclusions

Homotropanones with substituents at C-1 position can be accessed in a stereoselective fashion from δ-valerolactone. Formation of the Weinreb amide resulting from the nucleophilic opening of the lactone, followed by successive alcohol oxidation, *N*-*tert*-butanesulfinyl keto aldimine formation, decarboxylative Mannich reaction involving 3-oxobutanoic acid, and an intramolecular L-proline organocatalyzed Mannich cyclization, are the sequential steps in these transformations, which proceeded in fair to good yields. The stereochemical outcome is determined by the configuration of the sulfur atom of the sulfinyl group of the chiral sulfinyl imines, as it has been exemplified in the synthesis of both enantiomers of the adaline alkaloid. Unfortunately, tropanones could not be synthesized following the same sequence of reactions when starting from γ-butyrolactone, since the final L-proline organocatalyzed cyclization failed to achieve the desired compound.

## Data Availability

The data presented in this study are available in this article.

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
