# Peer review of "Stereoselective Synthesis of 1-Substituted Homotropanones, including Natural Alkaloid (−)-Adaline"

_molecules, 2023, doi:10.3390/molecules28052414_

Round 1

Reviewer 1 Report

1.     I suggest the authors do the optimization on the keywords.

2.     Could the authors propose the possible mechanism on such stereoselective synthesis?

3.     In the discussion section, the authors only list own stereoselective synthesis, why not extend and compare the similar methods?

4. “The current state of the research field should be carefully reviewed and key publications cited.” and “and the decarboxylative Mannich coupling of β-keto acids with of N-tert-sulfinyl imines, and the application of the resulting homoallyla-80 mine derivatives and β-amino ketones, respectively, to the synthesis of natural products.” This should be cited related refs, such as Org. Chem. Front., 2020,7, 3515-3520; New J. Chem., 2020, 44, 16265-16268; J. Org. Chem. 2019, 84, 14627−14635 and Org. Chem. Front., 2021, 8, 4554–4559

5. Source and purity of all chemicals used should be specified in the experimental section.

6. The manuscript contains spelling/grammatical errors. So, the language should be polished thoroughly. Such page 13, line 8.

Author Response

1. I suggest the authors do the optimization on the keywords.

Response: 

Former key words have been modified: chiral sulfinyl imines; decarboxylative Mannich reaction; diastereoselective additions; 9-azabicyclo[3.3.1]nonan-3-ones; homotropanone alkaloids; (Ë—)- and (+)-adaline.

New keywords: sulfinyl imines; decarboxylative Mannich reaction; homotropanone alkaloids, cyclizations; organocatalysis

2. Could the authors propose the possible mechanism on such stereoselective synthesis?

Response:

The stereoselectivity is achieved at the decarboxylative Mannich reaction of chiral keto aldimines 5 with 3-oxobutanoic acid to give amino diketone derivatives 6. This methodology was developed in our research group (reference [18]), and we commented in the main text about that “Concerning the stereochemical pathway, the nucleophilic addition of the enolate took place to the Re face of imine with SS configuration. This result was rationalized considering an eight-membered cyclic transition state A, supported also in DFT calculations (Scheme 5, Scheme 3 in the revised version)”. For the organocatalyzed cyclization to produce target homotropanones 7, a working model was proposed in Scheme 6 (Scheme 3 in the revised version), and to rationalized the racemization of these compounds 7, we proposed a mechanism which is depicted on Scheme 7 (Scheme 4 in the revised version).

3. In the discussion section, the authors only list own stereoselective synthesis, why not extend and compare the similar methods?

Response:

It is true, as the Reviewer said, that in the “Results and Discussion” section we only commented on our synthesis. Compared to other approaches to these 9-azabicyclo[3.3.1]nonan-3-ones, we said in the new paragraph above Scheme 2 (Retrosynthetic analysis) in the “Introduction” section that “…the synthetic strategy we have envisioned for the synthesis of these bicyclic homotropanones, which is closely related to the strategy reported by Davis [13] (Scheme 2).” We commented also what we consider to be significant synthesis of alkaloids adaline and euphococcinine in the paragraph before Scheme 1. In addition, we provided in reference [10] a review: “Recent progress in the synthesis of homotropane alkaloids adaline, euphococcinine and N-methyleuphococcinine. Beilstein J. Org. Chem. 2021, 17, 28–41.”

4. “The current state of the research field should be carefully reviewed and key publications cited.” and “and the decarboxylative Mannich coupling of β-keto acids with of N-tert-sulfinyl imines, and the application of the resulting homoallyla-80 mine derivatives and β-amino ketones, respectively, to the synthesis of natural products.” This should be cited related refs, such as Org. Chem. Front., 2020,7, 3515-3520; New J. Chem., 2020, 44, 16265-16268; J. Org. Chem. 2019, 84, 14627−14635 and Org. Chem. Front., 2021, 8, 4554–4559

Response:

These references are not related to the topic of this article.

5. Source and purity of all chemicals used should be specified in the experimental section.

Response:

The first sentence of “Materials and Methods” section has been modified according to the Reviewer’s comment: “Reagents and solvents were of reagent grade and purchased from commercial suppliers (Sigma-Aldrich, Fisher Scientific), and used as received.”

6. The manuscript contains spelling/grammatical errors. So, the language should be polished thoroughly. Such page 13, line 8.

Response:

We apologize again for these mistakes. We have made the corrections in the revised version of the manuscript.

Reviewer 2 Report

Reviewer’s Comments (Manuscript ID # molecules-2256947)

The manuscript reported by Foubelo and co-workers describes the “Stereoselective synthesis of 1-substituted homotropanones, including natural alkaloid (Ë—)-Adaline”. In current manuscript author used N-tert-butanesulfinyl imines as reaction intermediate for the  stereocontrolled synthesis of 1-substituted homotropanones. Futher, the utility of the method was demonstrated by a synthesis of natural product (-)-adaline and its enantiomer (+)-adaline. However, the generality of this synthetic methodology has shown by preparing very limited substrate (only 6) and also using this method tropanon derivative 16 was failed to prepare.

This manuscript needs major revision prior to publication in Molecules.

1.      In introduction section, first paragraph on page 2 (starts from page 1 line 43) looks like author copied the comments given by previous reviewer.

2.      Instead of making four different schemes (schemes 3-6), one scheme can be provided with general structure.  

3.      Generality of synthetic methodology can be generalized by synthesizing variety additional compounds.

4.      Compound 6a is cited in the test but missing in the scheme 5.

5.      The caption of scheme 8 is retrosynthetic analysis but in scheme 8, synthesis of compound 16 is provided.

6.      Some of the references are not uniform style.

7.      Manuscript needs to be checked for typo errors.

Author Response

The manuscript reported by Foubelo and co-workers describes the “Stereoselective synthesis of 1-substituted homotropanones, including natural alkaloid (Ë—)-Adaline”. In current manuscript author used N-tert-butanesulfinyl imines as reaction intermediate for the stereocontrolled synthesis of 1-substituted homotropanones. Futher, the utility of the method was demonstrated by a synthesis of natural product (-)-adaline and its enantiomer (+)-adaline. However, the generality of this synthetic methodology has shown by preparing very limited substrate (only 6) and also using this method tropanon derivative 16 was failed to prepare.

This manuscript needs major revision prior to publication in Molecules.

1. In introduction section, first paragraph on page 2 (starts from page 1 line 43) looks like author copied the comments given by previous reviewer.

Response:

We apologize for this error. Actually, the manuscript was written in a different format than the template, and then we copied the paragraphs and schemes, and placed them in the correct position in the template. We sent to Molecules by mistake what was not the final version of our manuscript. This paragraph is related to the recommendations of the journal template. We have removed it from the revised version. In addition, we added the above paragraph of Scheme 2 (Retrosynthetic analysis) in the “Introduction” section that was lost in the first version.

“Continuing our interest in the use of N-tert-butanesulfinyl imines as electrophiles [25-29], we decided to explore new synthetic pathways to access 1-substituted homotropanones in an enantioenriched form involving these chiral imines. Sequential decarboxylative Mannich reaction of a chiral N-tert-butanesulfinyl keto aldimine and a β-keto acid, followed by an organocatalyzed intramolecular Mannich reaction, upon desulfinylation, are key steps in the synthetic strategy we have envisioned for the synthesis of these bicyclic homotropanones, which is closely related to the strategy reported by Davis [13] (Scheme 2).”

2. Instead of making four different schemes (schemes 3-6), one scheme can be provided with general structure. 

Response:

Following indication of the reviewer, former Schemes 3-6 have been removed and a new Scheme 3 has been created containing all the information which was provided in former Schemes 3-6. As a consequence, former Schemes 7 and 8 are now 5 and 6, respectively.

On this point we disagree with the reviewer’s opinion. We would prefer to keep former Schemes 3-6 where the structures of all compounds are clearly represented, although the reviewer’s indication is addressed in the revised version (at the same time, we have prepared a revised version with formers Schemes 3-6).

3. Generality of synthetic methodology can be generalized by synthesizing variety additional compounds.

Response:

It is true that not many 1-substituted homotropanones were synthesized (only 8). Regarding the scope, we considered homotropanones with a linear hydrocarbon chain (7b, 7c and 7d), linear hydrocarbon chains with a terminal double bond (7e), and with an aromatic ring at 2- and 3-positions (7h and 7i). Homotropanones with phenyl and benzyl groups cannot be prepared, and this is a limitation of the methodology. On the other hand, in previous synthesis of 1-substituted homotropanones (references 10-14), authors reported the synthesis of just one or two derivatives with aliphatic groups (adaline: n-pentyl; euphococcinine: methyl). Furthermore, our straight forward method allowed access to both enantiomers.

4. Compound 6a is cited in the test but missing in the scheme 5.

Response:

Compound 6a is now on new Scheme 3 (<5% yield). It was not shown in former Scheme 5 because it could not be characterized, although as the reviewer pointed out, it was cited in the main text.

5. The caption of scheme 8 is retrosynthetic analysis but in scheme 8, synthesis of compound 16 is provided.

Response:

The reviewer is right and we apologize again for our mistake in not sending the final version of the manuscript. The caption of new Scheme 5 (former Scheme 8) is “An attempt to synthesize 1-substituted 8-azabicyclo[3.2.1]octan-3-one 16 from γ-butyrolactone 1b.”

6. Some of the references are not uniform style.

Response:

The title of articles in references 12, 13, 30, 31 and 32 are underlined. These titles were copied from the journal web page and the format of the manuscript word and pdf documents we sent to Molecules was correct. The underlined style was probably revealed after manipulation of these documents by the editorial office, due to the preferences of the former document on the original web page.

7. Manuscript needs to be checked for typo errors.

Response:

We apologize again for these mistakes. We have made the corrections in the revised version of the manuscript.

Round 2

Reviewer 1 Report

ok

Reviewer 2 Report

The author has addressed the most of concerns raised by reviewers, so this reviewer is recommending this manuscript for publication.